# Automatic Object Detection Algorithm-Based Braille Image Generation System for the Recognition of Real-Life Obstacles for Visually Impaired People

**DOI:** 10.3390/s22041601

**Published:** 2022-02-18

**Authors:** Dayeon Lee, Jinsoo Cho

**Affiliations:** IT Convergence Engineering and Computer Convergence Major, Gachon University, Seongnam 13120, Korea; lidy030@gachon.ac.kr

**Keywords:** image processing, object detection, artificial intelligence, blind, braille system

## Abstract

The global prevalence of visual impairment due to diseases and accidents continues to increase. Visually impaired individuals rely on their auditory and tactile senses to recognize surrounding objects. However, accessible public facilities such as tactile pavements and tactile signs are installed only in limited areas globally, and visually impaired individuals use assistive devices such as canes or guide dogs, which have limitations. In particular, the visually impaired are not equipped to face unexpected situations by themselves while walking. Therefore, these situations are becoming a great threat to the safety of the visually impaired. To solve this problem, this study proposes a living assistance system, which integrates object recognition, object extraction, outline generation, and braille conversion algorithms, that is applicable both indoors and outdoors. The smart glasses guide objects in real photos, and the user can detect the shape of the object through a braille pad. Moreover, we built a database containing 100 objects on the basis of a survey to select objects frequently used by visually impaired people in real life to construct the system. A performance evaluation, consisting of accuracy and usefulness evaluations, was conducted to assess the system. The former involved comparing the tactile image generated on the basis of braille data with the expected tactile image, while the latter confirmed the object extraction accuracy and conversion rate on the basis of the images of real-life situations. As a result, the living assistance system proposed in this study was found to be efficient and useful with an average accuracy of 85% a detection accuracy of 90% and higher, and an average braille conversion time of 6.6 s. Ten visually impaired individuals used the assistance system and were satisfied with its performance. Participants preferred tactile graphics that contained only the outline of the objects, over tactile graphics containing the full texture details.

## 1. Introduction

The leading cause of visual impairment can be congenital or a result of accidents, aging, or diseases. In addition, the number of people with acquired vision loss is increasing because of urban environmental factors resulting from the development of electronic devices [1,2]. A survey made by the World Health Organization (WHO) in 2020 indicated that approximately 2.2 billion people, which accounts for 28.22% of the global population, are visually impaired (i.e., near or distance visual impairment) [3,4].

Visually impaired people rely on their auditory perception and somatosensation—primarily sound and braille—to obtain information from the environment; they use assistive devices such as canes to recognize obstacles. However, although 28.22% of the global population accounts for visually impaired individuals [5,6], accessible facilities are not universally installed, leading to issues of social discrimination due to the limitations of their activities. Particularly, they cannot face unexpected situations outdoors independently, thereby restricting their activities to indoors or in their neighborhood. Accessible facilities such as tactile pavements and tactile signs are not appropriately installed in all institutions. Moreover, some countries do not provide support for assistive devices. In addition, most artworks, such as paintings and sculptures, cannot be touched to preserve them, making it difficult for visually impaired people to enjoy cultural activities through their imagination alone with tactile brochures. Therefore, researchers conducted numerous studies to help them become self-sufficient in their daily lives. In particular, studies on providing information via braille have recently gained attention. However, most of these studies focused on tactile maps or graphic image braille conversion. A system is needed worldwide to ease their daily lives because it is difficult to assist the visually impaired individuals in real life.

This study proposes a living assistance system based on images of the surroundings and objects that visually impaired people want to experience in real life that are captured by smart glasses. The system stores object information using an object detection algorithm to provide voice guidance when the user goes outdoors. Moreover, the system provides an object image braille conversion service using an object extraction algorithm when indoors and carrying a braille pad. The braille data are generated as binary data to enable use in various braille pads, and the images are generated at three degrees of expression to enable users to recognize the shapes at different types. The accuracy of the proposed system is calculated by comparing the example tactile image with the expected tactile image on the basis of the braille data, and the usefulness of the system is evaluated by comparing the object detection results in real-life images and the execution time.

## 2. Related Research

Researchers conducted various studies regarding the living assistance for visually impaired people. Previous studies were focused on the generation of tactile signs and maps as navigation aids for the visually impaired, image conversion, and the development of tactile image output devices for braille pads. However, there is a lack of studies on the generation of tactile images based on real-life images or systems that assist with real-life outdoor activities, such as the automatic object detection voice guidance system proposed in this study.

### 2.1. Similar Research

#### 2.1.1. Tactile Graphics

Tactile maps and images are generated through image processing based on general maps to create tactile maps. Tactile maps are the most provided navigation aid for the visually impaired people by public institutions. However, tactile maps are gradually being provided by various institutions, fueling further research on their development.

Kostopoulos et al. [7] proposed a method for generating tactile maps based on a map image created by reading the road names written on a map via OCR and converting it into a road image, as shown in Figure 1. Although the proposed system for creating tactile maps can quickly recognize roads on the basis of the road names, it cannot detect alleys without a name. Moreover, OCR is slow and limited although it is faster than the existing algorithms.

Zeng et al. [8] developed an interactive map in which the user can zoom in and out, as shown in Figure 2. They allowed users to explore the tactile map by dividing it into zoom levels. However, a post-experiment survey found that visually impaired people preferred maps with only two zoom levels, and the usage time increased due to various factors such as the production of the interactive map, the guidance of the selections, and the selection.

Moreover, Krufkaf et al. [9] proposed an advanced braille conversion algorithm for vector graphics on the basis of previous studies. The algorithm extracted object boundaries using the outline information of the graphic based on the vector graphics hierarchical characteristics. The levels are classified on the basis of the extracted boundaries, and the multi-level braille is converted to a braille tablet using the tiger advantage braille printer program [10]. Although the proposed multi-level braille conversion system can provide meaningful results, it is difficult to apply to real-life objects using vector graphics, as shown in Figure 3.

In Korea, Kim et al. [11] investigated braille conversion on the basis of images captured via a webcam. The locations with and without data are compared to identify characters in the image by analyzing the images using MATLAB. Figure 4 shows the evaluation of the recognition level according to the font size, font type, and camera performance. In addition, an algorithm was developed by configuring an optimal environment based on the evaluation results. Although their research showed significant results, the system can only convert numbers and uppercase English letters, and it cannot identify objects other than letters or recognize Korean letters.

Lee et al. [12] developed a banknote recognition system using Raspberry Pi as a camera. The process consisted of two steps (i.e., extraction and matching). The researchers compared the extraction algorithms SIFT, SURF, and ORB; they adopted SIFT because it yielded the highest recognition rate. The system achieved high accuracy even when changing the shooting method or in unsuitable environments (e.g., low light or rotated banknote) by generating vector images using extreme values as features. Nevertheless, the brute-force algorithm requires extensive time for recognition, as shown in Figure 5, making it unsuitable for this study, which uses many objects.

#### 2.1.2. Braille Pad

Researchers have made several attempts to output tactile images by combining a haptic device with a braille display [13].

Kim, S. et al. [14,15] proposed a 2D braille display to output data in the digital accessible information system (DAISY) and the electronic publication (EPUB) formats. They developed the braille pad for outputting braille information and the technology for tactile image conversion, as shown in Figure 6. Tactile image tests were conducted using simulators, and the tactile image conversion technology quantizes and binarizes data to convert graphs, graphic images, and even photos, enabling them to obtain significant results.

Prescher et al. [16] proposed a PDF-editor-based braille pad and braille conversion system. The user interface (UI) for displaying and editing PDF content was designed to show on one screen using a horizontally long touch-enabled braille pad. As both the content and editing UI are displayed on one screen, excessive information is provided at once, making it difficult for first-time users. Moreover, it can only translate the diagrams and text input, which are in PDF files rather than images, although it can display diagrams as shown in Figure 7.

#### 2.1.3. Supplementation and Service

In addition, various products and services are being researched to assist the visually impaired.

Kłopotowska et al. [17] studied architectural typhlographics and developed them through multi-criteria analysis by integrating the characteristics of braille maps and architectures (Figure 8). The study results show the future growth potential of typhlographics on the basis of its social values of enabling tourism for the visually impaired in addition to its broad utility in the development of tactile architectural drawings such as diversification of architectural education and interior design.

Morad [18] studied the assistive devices that receive location coordinates via the global positioning system (GPS) and process data through a PIC controller to output specific voice messages stored in the device for visually impaired people. The study aimed to develop an affordable and easy-to-use assistive device that helps the visually impaired people find their way on their own as they listen to the voice messages through the headset. It received a positive response from them when the device was used by people with visual impairments.

On the other hand, Fernandes et al. [19] proposed a radiofrequency identification (RFID)-based cane navigation system to guide people with visual impairments by using the RFID device installed under the road. The navigation system provides audio navigation assistance to reach the desired destination through the route calculation and location tracking using the RFID tags once the user inputs a specific destination in the cane. It is considered to have a significant growth potential owing to its higher accuracy than GPS and the easy-to-update feature of the navigation system.

Liao et al. [20] proposed the integration of the GPS and RFID technologies to develop a system for indoor use in order to address the shortcoming of the GPS system used. This hybrid system receives location data based on GPS and fine tunes the specific location data with RFID, which was developed to provide walking assistance to users. The study results are expected to facilitate the development of the walking assistance system for the visually impaired individuals and the enhancement of GPS accuracy.

### 2.2. Algorithms

#### 2.2.1. YOLO

You Only Look Once v3(YOLOv3), a Darknet-53 network-based object detection algorithm, passes through layers of various sizes and compares them with object characteristics analyzed in the dataset to detect objects [21,22,23]. This study used YOLOv3 for object detection to identify objects within the line of sight of users. YOLOv3 has undergone several versions of development, making it more accurate than other algorithms [24,25,26,27]. In addition, it is fast and specialized for real-time detection as it searches only once, enabling an object detection from images in real time. According to the study of Redmon et al. [23]. YOLOv3 yielded an mAP of 57.9% in a COCO dataset test, demonstrating the high speed and accuracy of the algorithm. Figure 9 and Figure 10 shows the YOLOv3 operating structure and the network structure, respectively. The method detected through the network is shown in Figure 11 and is expressed by Equation (Equation 1).
(1)$$\begin{array}{cc} b_{x} & =\sigma \left(t_{x}\right)+c_{x}\\ b_{y} & =\sigma \left(t_{y}\right)+c_{y}\\ b_{w} & =p_{w}e^{t_{w}}\\ b_{h} & =p_{h}e^{t_{h}} \end{array}$$

#### 2.2.2. Grabcut

The GrabCut algorithm allows more effective object feature classification and ease of use than previous algorithms [28,29], such as Magic Wand, Intelligent Scissors, Bayes Matte, Knockout2, and GraphCut. This algorithm is used to separate the detected objects from the background, exploiting its advantages of high speed and extraction accuracy with only user-specified regions. Through GraphCut-based segmentation, the color values between pixels are calculated. A color model is generated on the basis of the color values of the model, and the foreground and background are separated via segmentation, as shown in Figure 12. After adding a mask to distinguish the foreground and background on the basis of the selection of the user, the separated foreground can be re-extracted, as shown in Figure 13.

#### 2.2.3. Canny

In contrast to Contour, which is a contour line detection algorithm that generates boundary lines based on the height of the boundary detection target [30], Canny identifies the boundary values of the object to generate an outline [31]. In comparison to previous algorithms for generating outlines, Canny is fast and applicable to color images. Therefore, it was used to generate outlines for converting the extracted object to braille. In addition, new criteria were added to prevent it from generating abnormal outlines to achieve a low error rate and stable and improved system performance. Additionally, the criteria of existing algorithms are strengthened, and a parametric closed outline generation technique is provided through numerical optimization. Accordingly, additional criteria were hypothesized, and various equations and operators were used to satisfy the hypotheses. Figure 14 shows the results of this application, indicating its suitability as an outline generation algorithm.

## 3. System Design and Configuration of Use Environments

The automatic-object-detection-algorithm-based braille conversion system for the living assistance of the visually impaired mainly targets visually impaired people including those with limited sight who typically use braille since the system is fully operated by smartphones. The images of surrounding environment and objects are captured with smart glasses, and the braille images are generated on the braille pads. The relevant objects are captured through smart glasses, and the tactile image is the output on a braille pad. Figure 15 shows the structure of the system, which is operated through a smartphone. To detect objects, it is connected to smart glasses via Bluetooth using the smartphone. The camera screen of the smart glasses and the screen of the desired field of view are confirmed through the smartphone and a shooting request is sent when the smart glasses are connected. When the shooting request reaches the smart glasses, it takes a photo with the built-in camera and sends it to the smartphone. The location and name of the objects in the photo are transmitted through the smart glasses and confirmed via TTS when the system performs object detection at the request of the user. The image is converted to braille, and the braille data are transmitted to the braille pad to allow the user to confirm the shape of the object. Once the transmission is completed, the user can recognize the shape of the object with the tactile image generated through the braille pad.

## 4. System Configuration

Table 1 shows the configuration of the proposed system in five steps: shooting, object detection, object extraction, outline generation, and braille conversion. The algorithms for all steps except shooting are constructed on an integrated server to increase the processing speed and store and use various image data. Each step can be separately executed through a smartphone on the basis of the scope of use and selections of the user. Moreover, only the result data are stored on the smartphone. The data from each step are maintained until the step is executed again. Figure 16 presents the overall process of the system.

### 4.1. Object Detection

In the object detection step, the YOLOv3 algorithm was used to detect a variety of objects in real time. Figure 17 shows the results from the application of the system to a real object.

### 4.2. Object Extraction

The extraction step was configured using Python, and image processing algorithms used were from OpenCV. The objects were extracted using GrabCut after preprocessing the image. Figure 18 shows the structure of the object extraction step.

#### 4.2.1. Image Preprocessing

The contrast of the entire image, which refers to the difference in brightness between bright and dark areas in an image, is enhanced to clearly distinguish the colors of the detected image. An image with a small difference in brightness between bright and dark areas has a low contrast value, while an image with a large difference in brightness between bright and dark areas has a high contrast value. The contrast value refers to the contrast ratio. To increase the contrast value, dark areas must be darkened by increasing the color value of the pixels, and bright areas must be brightened by lowering the color values of the pixels.

Although there are various algorithms for increasing contrast value, the most basic technique is to multiply each pixel by a value based on the desired brightness of 1.0 [33,34]. Multiplication techniques are categorized into two methods: multiplying a MAT and using the saturate equation through the clip algorithm. However, they are not suitable for this study because these methods are mainly used on grayscale images to adjust only the brightness values. Instead, we examined algorithms used for colored images. The contrast of colored images is adjusted using a histogram equalization algorithm [35]. In addition, histogram smoothing converts a colored image composed of RGB channels into YCrCb channels and separates them into individual Y, Cr, and Cb channels, respectively, as shown in Figure 19. Y represents the luminance component, while Cr and Cb represent the chrominance components. Histogram equalization is applied to the separated luminance channels to increase the contrast value of the image.

Histogram equalization can be applied to an image composed of RGB channels to increase the contrast of the image. It increases the contrast by converting a colored image composed of RGB channels to YCrCb channels and separating them into individual Y, Cr, and Cb channels, respectively. Y represents the luminance component, while Cr and Cb represent the chrominance components. The contrast of color images is increased by applying the histogram equalization in the separated luminance component.

However, histogram equalization adjusts the contrast value of the entire image at once, making the bright areas very bright and dark areas very dark. This results in an unbalanced image overall. The CLAHE algorithm, which separately adjusts the brightness of specific areas in the image, was used to adjust the average brightness while increasing the contrast value [36]. To apply CLAHE, the image is converted to the LAB format and separated into individual channels to separate it into colored and grayscale [37] images. Channel L represents the brightness of the light and is expressed as a black and white image, while channels A and B represent the degree of color. Channel A represents magenta and green, and channel B represents blue and yellow. Moreover, the images are sequentially searched on the basis of the specified grid size, and the contrast value is adjusted to increase the contrast value in channel L and the black and white images. The channels are combined and converted back to the RGB format for other image processing after searching all images and adjusting the contrast value. Using the image from Figure 17, the contrast of an image was increased (Figure 20) through image channel separation, as shown in Figure 21.

#### 4.2.2. Object Extraction

The stored object location information is imported to extract objects from the image whose contrast was increased in the preprocessing step. Approximately 10 is added to or subtracted from each x and y value in the stored object location information to distinguish the surrounding pixels easily, as shown in Figure 22. GrabCut is used for object extraction. A black background is generated around it when an object is extracted, leaving only the object. In addition, the image size is reduced on the basis of the location information to fit the image size to the object and save it. Figure 23 shows the result of using the GrabCut algorithm.

### 4.3. Outline Generation

It is hard for users who have difficulty distinguishing objects to recognize objects with large amounts of information at once. Therefore, the tactile image generation was divided into three types depending on the desired type of expression of the user. These three types were “Out,” which displays only the outermost part such that the user can recognize the overall shape of the object; “Feature,” to ensure that the user can recognize the inner boundaries and form of the object; and “Detail,” which displays all information even the text in the object. Figure 24 shows the structure of the outline generation step.

#### 4.3.1. Image Processing

In the image processing step for generating the outline, the noise was removed and the colors were averaged. GaussianBlur was performed to remove the noise created by increasing the contrast and other noise [38]. GaussianBlur is used to remove large noise, while averaging [39,40] removes small components and detailed features, such as letters and shapes. Each algorithm was performed with varying degrees of frequency and intensity depending on the outline generation type selected by the user.

In the Out mode, starting from the 7 × 7 kernel and sigma 0, the algorithm was run as it gradually reduces the search size to ensure an iterative and powerful preprocessing and to completely remove noise, features, and information. In the Feature mode, starting from the 5 × 5 kernel and sigma 0, the algorithm was run as it gradually reduces the search size to moderately remove noise and information. On the other hand, in the Detail mode, it searched with a 3 × 3 kernel and sigma 0 to remove noise while maintaining features and information. Figure 25 shows the image processing results.

#### 4.3.2. Outline Generation

Canny [31] was used because it had a higher speed than Contour [30] although both Contour and Canny yielded similar accuracies for the outline generation algorithm. To generate the outline for each mode, the arrangement average and standard deviation of the images are calculated based on the noise-removed image, and the sum is set to a maximum value, so that the outline generation degree varies depending on the value range and mode. The morphology operations erosion and dilation were used to remove noise and small outlines remaining in the generated outline image. For braille conversion, the thickness was increased three times to confirm the line region, and the generated outlines were stored as individual images according to the mode. The thickness was increased three-fold by repeating the morphology dilation operation [41] three times, and the generated outlines were saved as an individual image on the basis of the mode to clearly define the lines for braille conversion. Figure 26 shows the result of outline generation.

### 4.4. Braille Conversion

Finally, the braille data were generated in the braille conversion step. For the data size, an image with a horizontal or vertical size of 416 was used as an input in the detection network in YOLOv3. The transformed image was resized on the basis of the detected location information of the object in the object detection process. Moreover, the data size was converted through braille data resizing on the basis of the received braille pad resolution when the braille pad was connected, ensuring that the output braille fitted the braille pad.

#### n × n Comparison Conversion

At this stage, the outline images generated through Python were imported and converted to braille data to create braille images. Colored values in which the color and brightness can be identified were searched via array comparison because images in Python are expressed as an array. A two-dimensional array of the same size as the image was generated to perform the search. The image was searched in a 5 × 5 pixel neighborhood, and it was checked whether there are color data in the center pixel (>0), as shown in Figure 27. A value of 1 was stored in the same location as the generated two-dimensional array if there are color data. An array was finally generated by searching the entire image, which is stored for transmission to the braille pad. The tactile image was generated by the same technique; the image was searched, and a circle was created in areas with a value. Figure 28 shows the results of braille transformation through comparative transformation.

## 5. Experiment and Evaluation

We evaluated the accuracy and usefulness of the tactile image generated by the proposed system. To evaluate the accuracy, the expected result images and the system result images for a variety of objects were compared. On the other hand, to evaluate the usefulness, the execution time of the system was calculated using photos in diverse situations that can be confirmed in real life, which verified the applicability of the system in real life.

### 5.1. Experiment

#### 5.1.1. Object Data Generation

The highly well-known and stable Microsoft COCO dataset [42] was used as the basic dataset because the dataset was required for object detection through YOLOv3. Table 2 lists the selected objects. Additionally, based on the COCO dataset object list, objects that give visually impaired people discomfort were added according to survey results, thus forming a dataset with 100 types of objects. The survey was conducted among visually impaired people in Korea at a welfare center. Table 3 summarizes the results.

#### 5.1.2. Accuracy Evaluation

To select the objects for the evaluation criteria, the objects that the visually impaired frequently use or encounter in real life were categorized into the following: (1) “indoors” and “outdoors” and (2) based on their sizes (i.e., large, medium, and small), resulting in a total of six objects. The following size criteria were applied: objects difficult to hold in the hands were classified as large, objects that can be held with two hands as medium, and objects that can be held with one hand as small. For the objects that are most frequently encountered outdoors, “car” was selected for large, “fire hydrant” for medium, and “traffic cone” for small. On the other hand, “closet” was selected for large, “chair” for medium, and “comb” for small for the objects that are most frequently used indoors. Figure 29 and Figure 30 show the comparison between the expected data and actual object results. The actual results were compared with [43,44] the braille for the “Detail” mode to verify the expression of details in the images.

#### 5.1.3. Usefulness Evaluation

To evaluate the usefulness, based on the three photos with the themes of “walking,” “eating,” and “washing face,” the conversion time in each step was measured and averaged, and the identified objects were compared with [43,44] the detected object list. Only the name and location value of the object closest to the user were used when there were duplicate objects in the braille conversion step, thus performing braille conversion without any duplicate objects. Figure 31 shows the photos used for the evaluation, converted photos, and detected objects list, with conversion times of 5.8, 4.5, and 7.4 s, respectively.

### 5.2. Overall Evaluation

The main object was compared with the expected generated data to evaluate the accuracy of the tactile image. We verified the amount of time needed for conversion to evaluate the usefulness of the system.

In the accuracy evaluation, the expected result image was visually compared with the resulting image of the system, and the accuracy of the generated tactile image was measured. The results showed that the final image has an average accuracy of 85% which is similar to that of the expected image.

In the usefulness evaluation, the list of detected objects was compared and the conversion time was measured on the basis of the photos of three situations that users can encounter in real life. For the objects detected in photos of real-life situations, the results indicated an accuracy of approximately > 90%. By excluding duplicate objects, the average time needed to convert the objects was less than 6.6 s, exhibiting that it can be quickly used in real life.

This system can output tactile images generated on the basis of braille data of objects with a shape similar to those of real-life objects, yielding significant results.

Ten visually impaired individuals were satisfied with the performance of the assistance system. Moreover, they preferred the Out type, which simplifies the tactile information in a straightforward manner, over the Detail type, which converts the real objects of complex composition.

## 6. Conclusions and Discussion

### 6.1. Conclusions

The proposed system was designed to inform visually impaired people about the types of obstacles in their field of view and to help them recognize their shapes. The system used an AI algorithm with high processing speed to quickly guide the user and integrated a simple image processing algorithm to provide tactile images in a short time. This study proposes a new and simple type of assistive device for visually impaired people who usually use braille, including people with limited sight. However, new algorithms or the latest technologies were not applied in the proposed system. The proposed braille conversion algorithm yielded an accuracy of 85% in relation to the expected result, demonstrating its usefulness. By excluding duplicate objects, approximately 12 out of 13 objects that can be confirmed in real life were detected on average. In addition, the conversion took an average of 6.6 s, indicating that the system is sufficient for use in real life.

### 6.2. Discussion

This study proposes a living assistance system that is applicable both indoors and outdoors by integrating object recognition, object extraction, outline generation, and braille conversion algorithms. According to the experiments and evaluations, we found that the system developed on the basis of the database tailor-made to the needs of visually impaired people (includes people with limited sight), who usually use braille, was useful.

However, some limitations of this study include the object extraction results obtained through GrabCut using the coordinates of the detected objects with YOLOv3 did not match with the real object. Moreover, some images other than the object image are left, indicating an inaccuracy in the braille conversion.

Therefore, we plan to perform primary development research to further improve the accuracy of the system and to generate and apply YOLOv3-based object masks although a more advanced system may require additional conversion time. Furthermore, we plan to conduct secondary development research to convert detected objects to icons and reflect the areas of improvement found from tests.

## Figures and Tables

**Figure 1 sensors-22-01601-f001:**
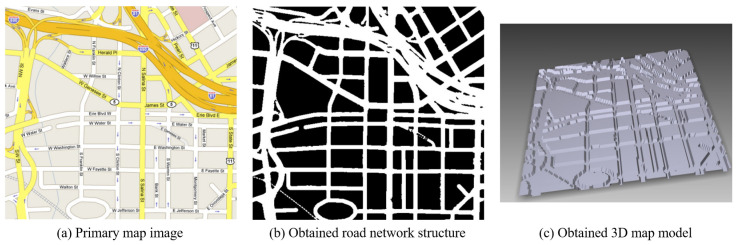
Map image-based tactile map production method [7].

**Figure 2 sensors-22-01601-f002:**
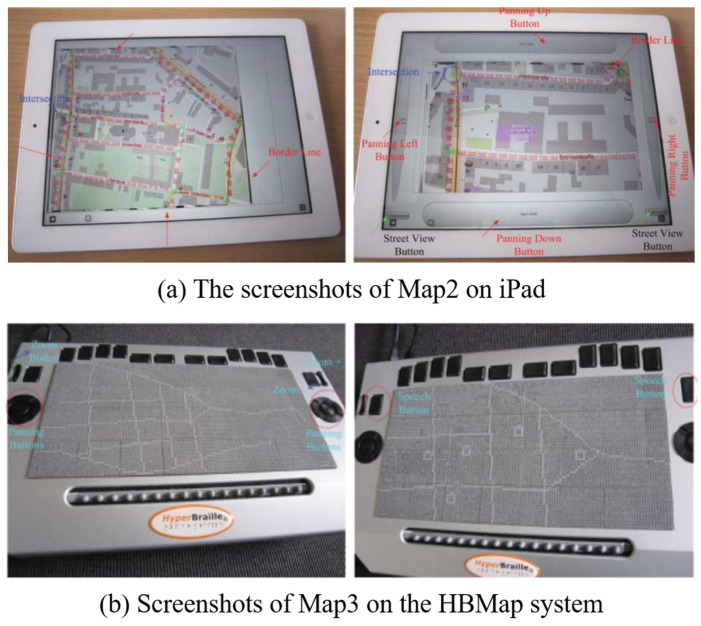
iPad and HBMap system-based interactive maps [8].

**Figure 3 sensors-22-01601-f003:**
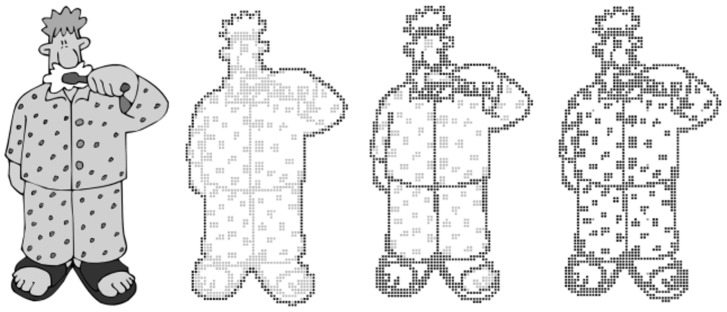
Outputs of proposed method for the vector graphic [9].

**Figure 4 sensors-22-01601-f004:**
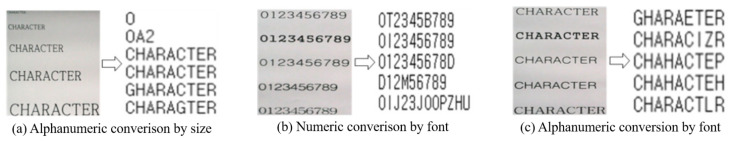
Image conversion according to font size and font [11].

**Figure 5 sensors-22-01601-f005:**
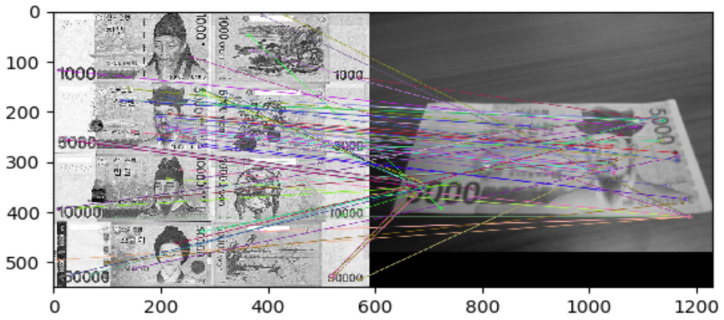
Keypoints Matching Using the Brute-Force Algorithm [12].

**Figure 6 sensors-22-01601-f006:**
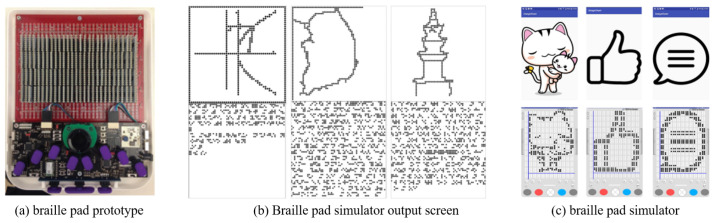
Braille pad prototype and Output screen [14,15].

**Figure 7 sensors-22-01601-f007:**
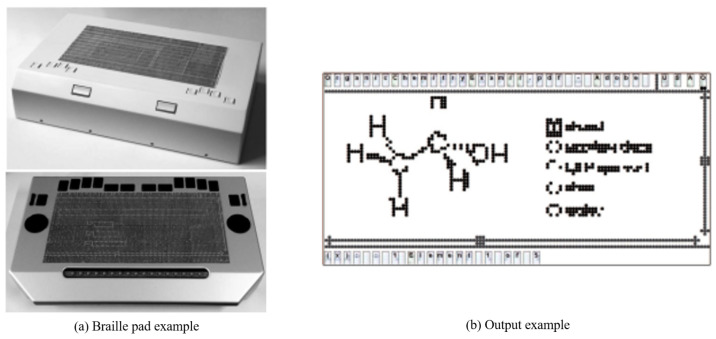
Braille pad and Output example [16].

**Figure 8 sensors-22-01601-f008:**
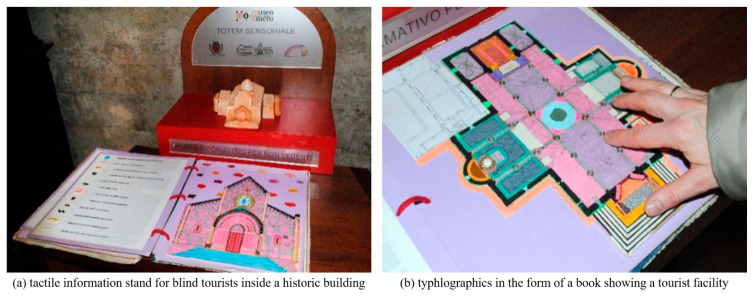
Typhlogics in the form of a book that shows tactile information tables and tourist facilities for blind tourists [17].

**Figure 9 sensors-22-01601-f009:**
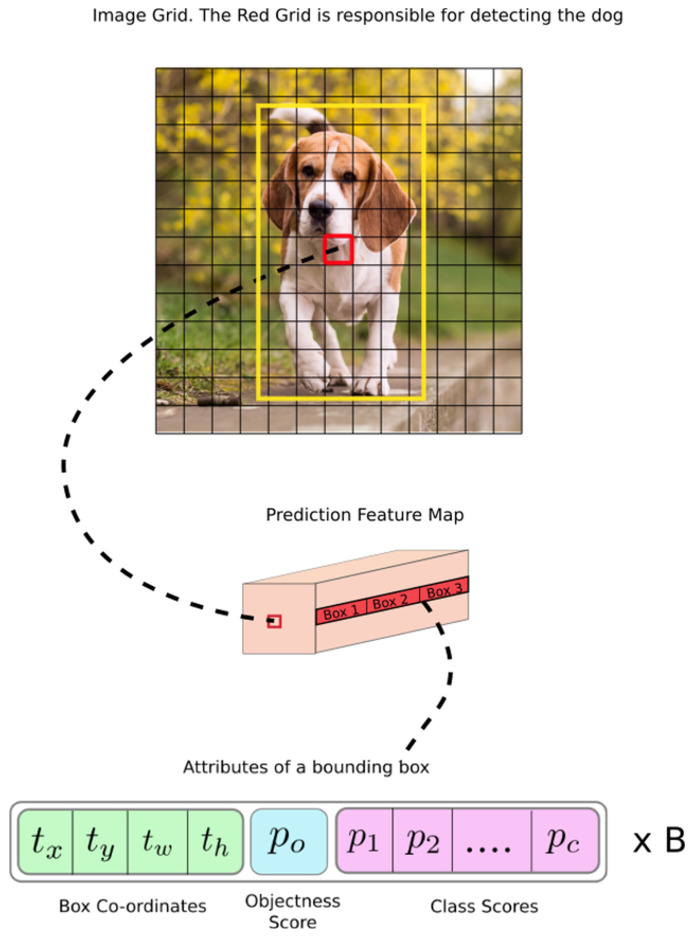
YOLOv3 network detection method [24].

**Figure 10 sensors-22-01601-f010:**
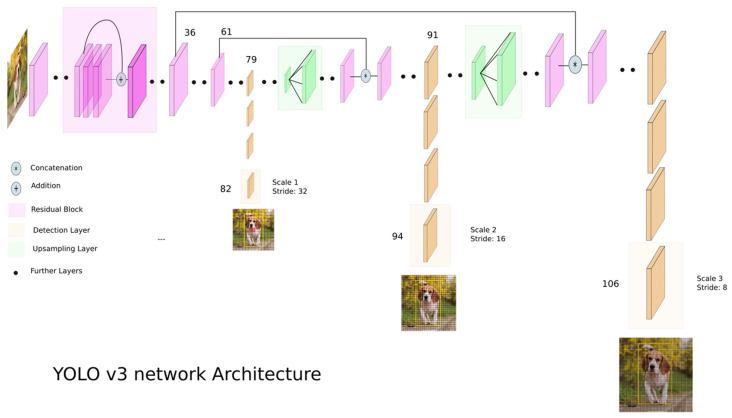
YOLOv3 network architecture [24].

**Figure 11 sensors-22-01601-f011:**
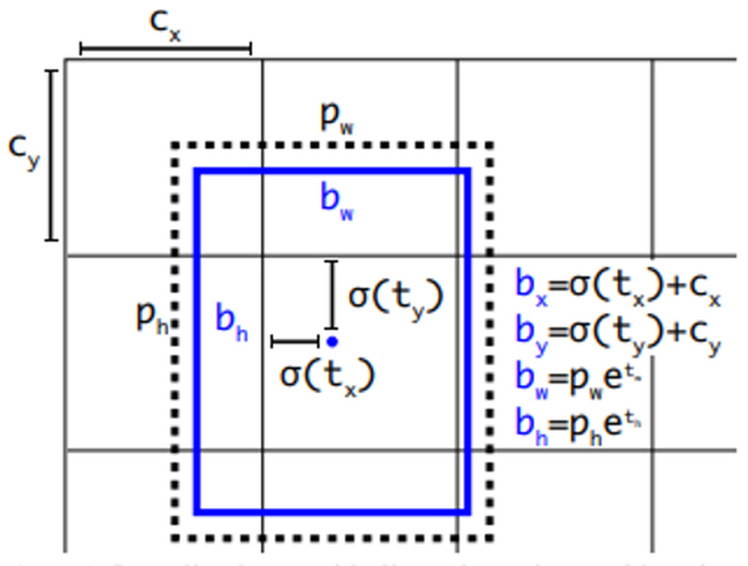
Numerical expression of YOLOv3 object detection [23].

**Figure 12 sensors-22-01601-f012:**
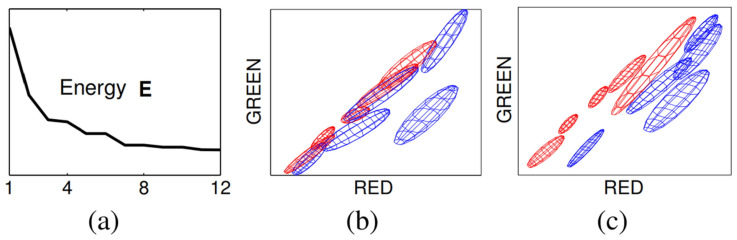
GrabCut principle image-Convergence of iterative minimization. (**a**) The energy E for the llama example converges over 12 iterations. The GMM in RGB colour space (side-view showing R,G) at initialization (**b**) and after convergence (**c**) [28].

**Figure 13 sensors-22-01601-f013:**
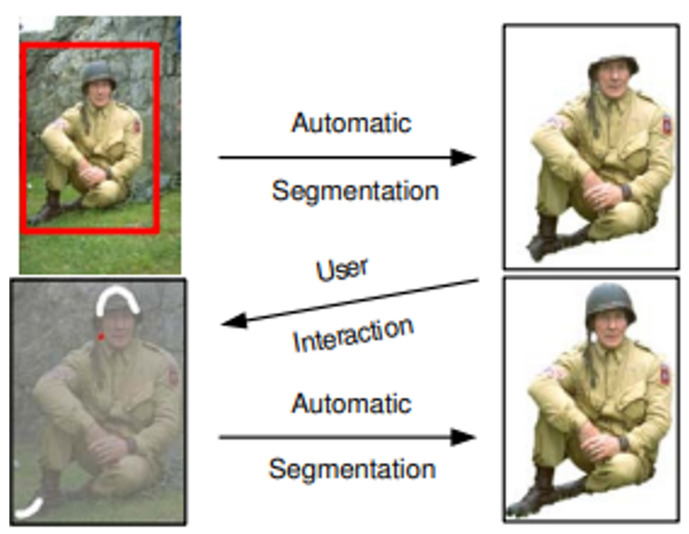
Grabcut example image [28].

**Figure 14 sensors-22-01601-f014:**
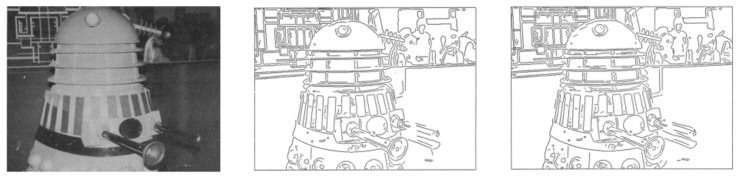
Canny example image [31].

**Figure 15 sensors-22-01601-f015:**
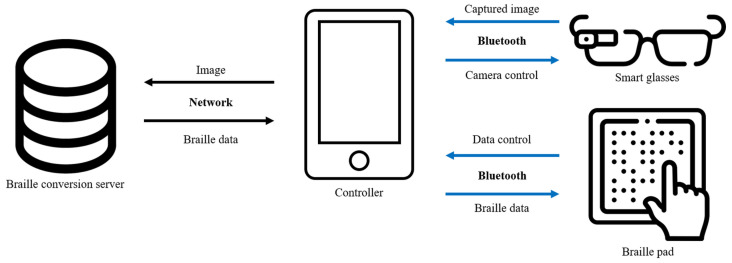
System schematic.

**Figure 16 sensors-22-01601-f016:**
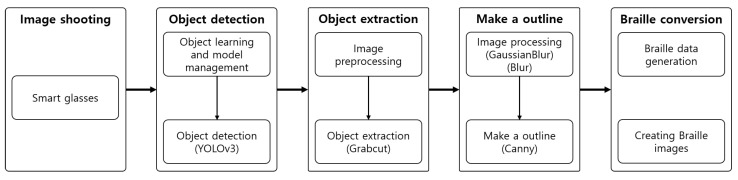
Process.

**Figure 17 sensors-22-01601-f017:**
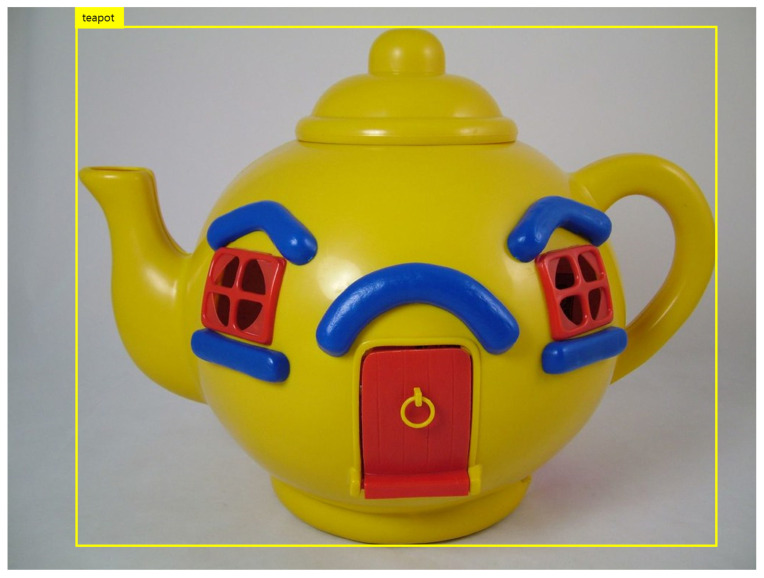
Object detection example [32].

**Figure 18 sensors-22-01601-f018:**
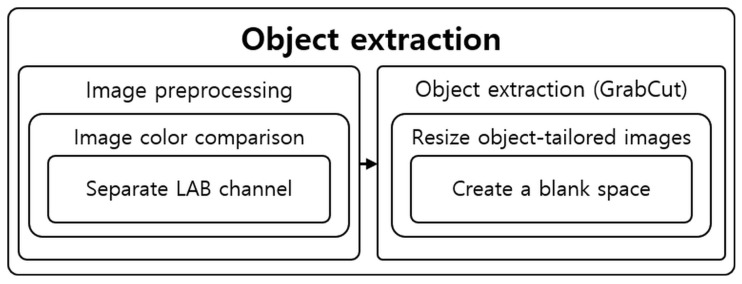
Structural diagram of object extraction steps.

**Figure 19 sensors-22-01601-f019:**
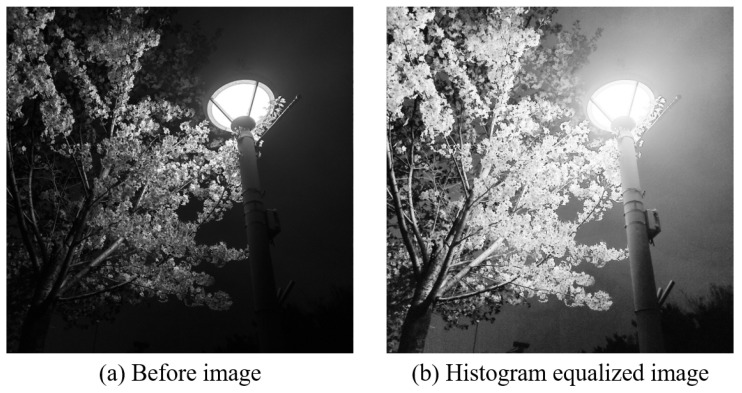
Histogram equalization Example.

**Figure 20 sensors-22-01601-f020:**
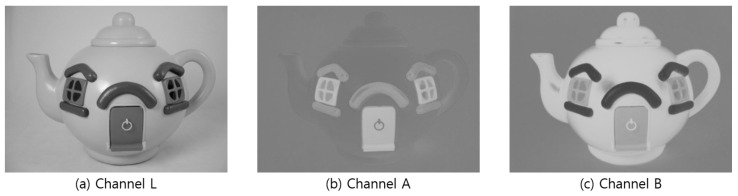
LAB Image by channel.

**Figure 21 sensors-22-01601-f021:**
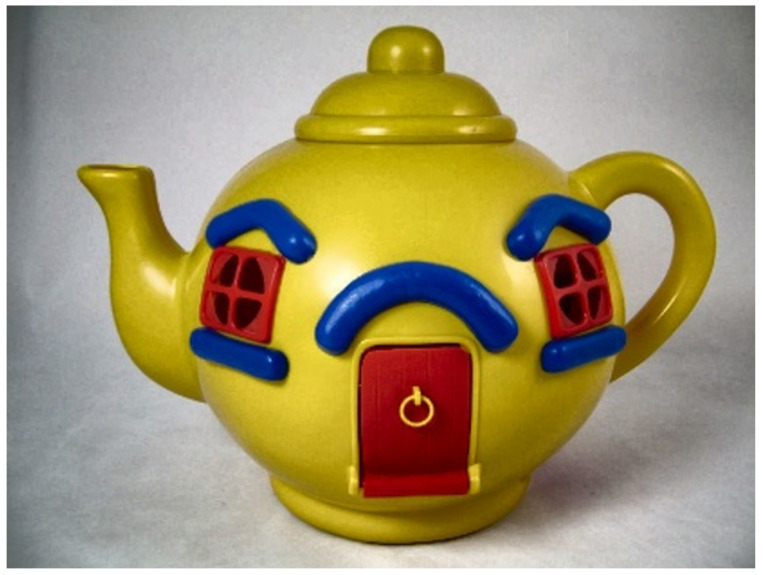
Contrasted image.

**Figure 22 sensors-22-01601-f022:**
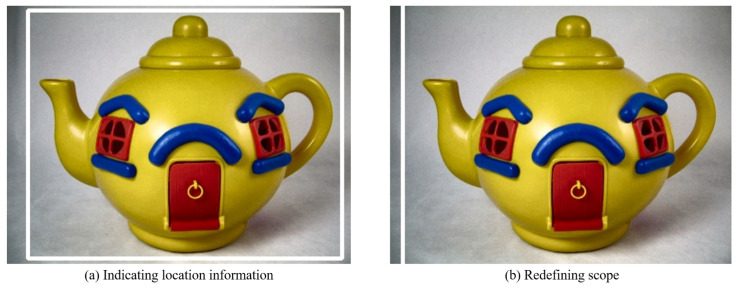
Object measurement range.

**Figure 23 sensors-22-01601-f023:**
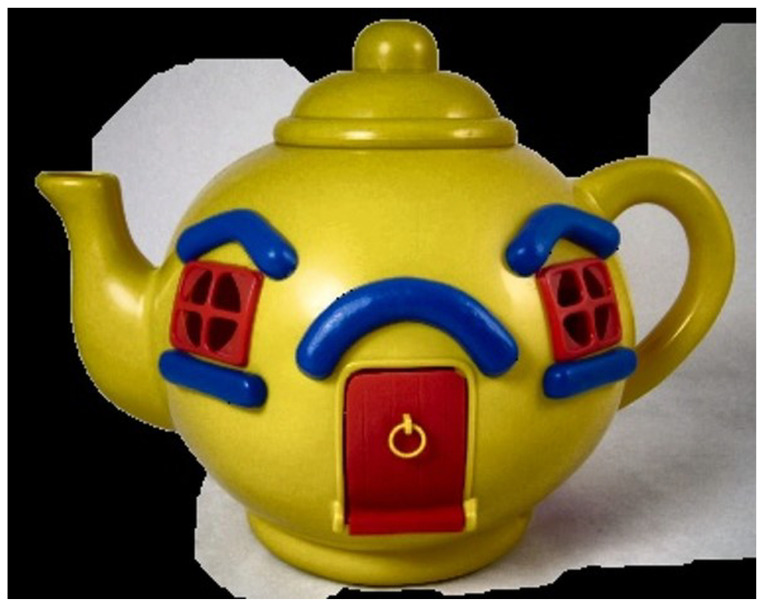
Object extracted image.

**Figure 24 sensors-22-01601-f024:**
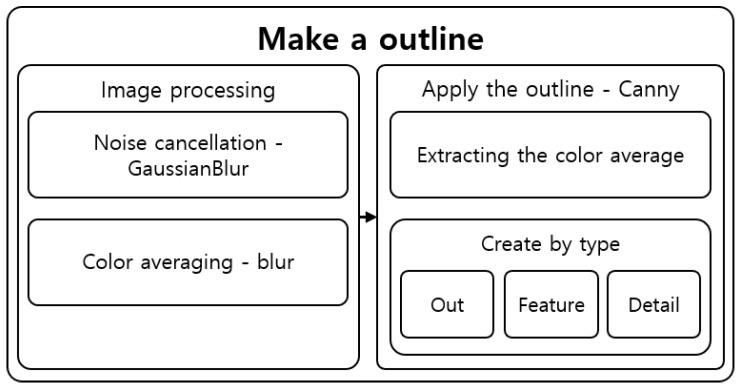
Outline creation step structure.

**Figure 25 sensors-22-01601-f025:**
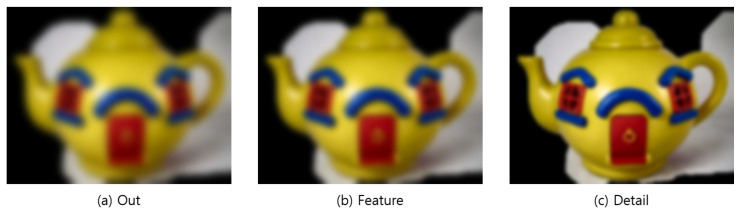
Image after performing.

**Figure 26 sensors-22-01601-f026:**
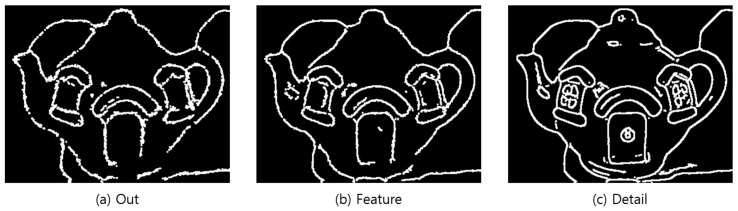
Outline image—Step-by-step image completed up to thickness increase.

**Figure 27 sensors-22-01601-f027:**
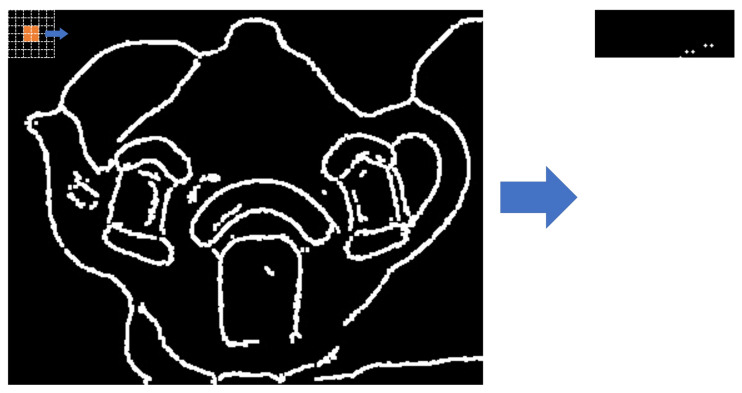
Example of braille conversion process.

**Figure 28 sensors-22-01601-f028:**
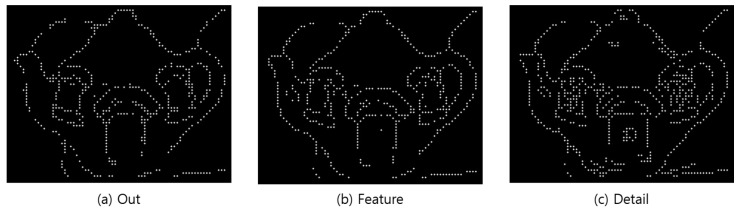
Generated braille image.

**Figure 29 sensors-22-01601-f029:**
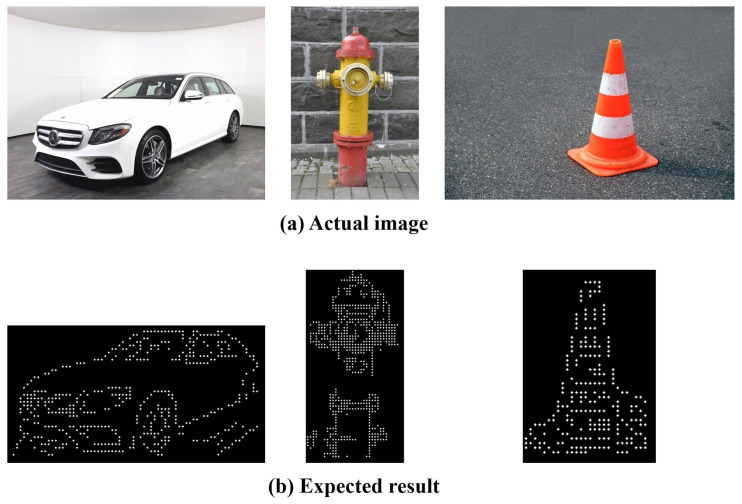
Comparison of expected and actual data(Outdoors).

**Figure 30 sensors-22-01601-f030:**
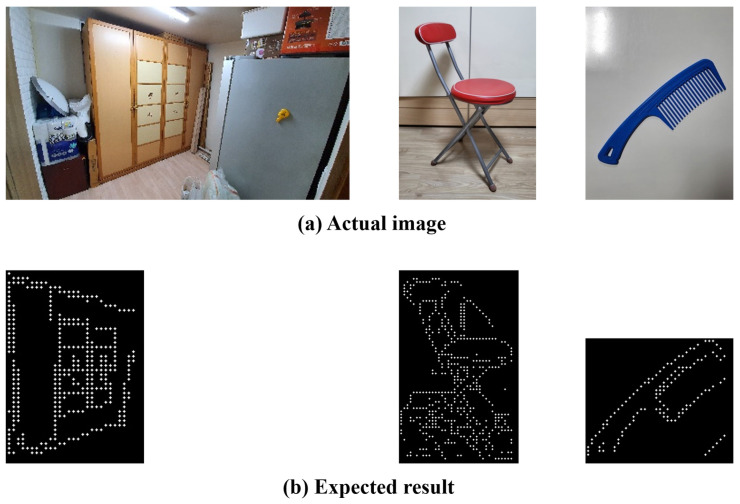
Comparison of expected and actual data(Indoors).

**Figure 31 sensors-22-01601-f031:**
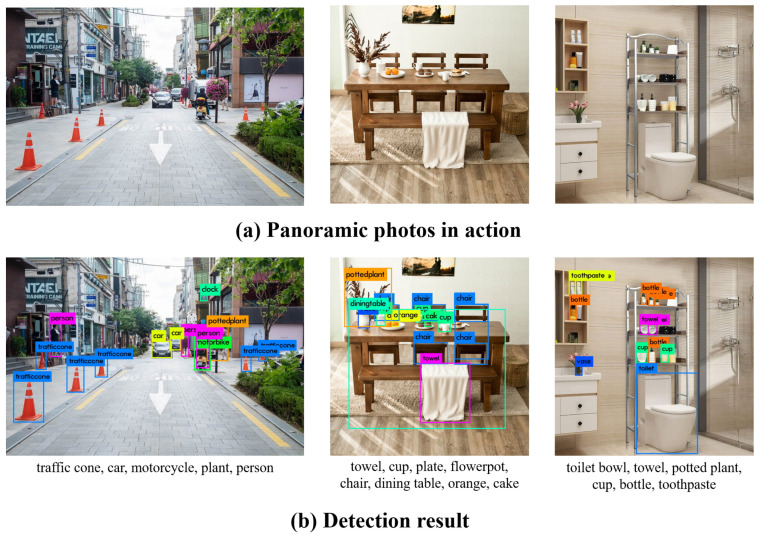
As a result of applying it to real life photos.

**Table 1 sensors-22-01601-t001:** Requirements of proposed system.

Function	Description
Image shooting	Capture photo of user-specified field of view and generate image
	Transfer to image controller and store
	Receive voice guidance data at user request
Object detection	Learn object images in database defined by system administrator
	Generate object recognition model
	Recognize objects based on image and store result image
	Store analysis result data
	Transmit voice guidance data at user request
Object extraction	Extract objects from image based on data
	Resize and store extracted object images
Outline generation	Preprocess image
	Calculate average color values based on extracted object images
	Generate object outline based on color values
Braille conversion	Analyze generated outline and create braille data
	Analyze resolution of linked braille pad
	Convert data size to braille pad resolution

**Table 2 sensors-22-01601-t002:** COCO dataset object list [42].

Person	Backpack	Umbrella	Handbag	Tie	Suitcase	Bicycle	Car	Motorcycle	Airplane
Bus	Train	Truck	Boat	Traffic light	Fire hydrant	Stop sign	Parking meter	Bench	Bird
Cat	Dog	Goose	Sheep	Cow	Elephant	Bear	Zebra	Giraffe	Frisbee
Skis	Snowboard	Sports ball	Kite	Baseball bat	Baseball glove	Skateboard	Surfboard	tennis racket	Bottle
Wine glass	Cup	Fork	Knife	Spoon	Bowl	Banana	Apple	Sandwich	Orange
Broccoli	Carrot	Hot dog	Pizza	Donut	Cake	Chair	Couch	Potted plant	Bed
Dining table	Toilet	TV	Laptop	Mouse	Remote	Keyboard	Cell phone	Microwave	Oven
Toaster	Sink	Refrigerator	Book	Clock	Vase	Scissors	Teddy bear	Hair drier	Toothbrush

**Table 3 sensors-22-01601-t003:** List of selected objects.

Person	Backpack	Umbrella	Handbag	Tie	Suitcase	Bicycle	Car	Motorcycle	Airplane
Bus	Train	Truck	Traffic light	Fire hydrant	Subway	Bench	Bird	Cat	Dog
Sports ball	Skateboard	Bottle	Wind glass	Cup	Fork	Knife	Spoon	Bowl	Chair
Tissu	Potted plant	Bed	Dining table	Toilet	TV	Laptop	Mouse	Remote	Keyboard
Cell phone	Microwave	Sink	Refrigerator	Book	Clock	Pillow	Scissors	Toothbrush	Toothpaste
Hair drier	Braille pad	Tree	Street lamp	Utility pole	Manhole	Vending machine	Elevator	Standing board	Escalator
Shampoo	Conditioner	Lottion	Stair	Traffic cone	Bollard	Radio	Desk	Whellchair	Eletric rice cooker
Gas cooker	Closet	Washing machine	Teapot	Electric fan	Comb	Bookmark	Soap	Glasses	Key
Shoes	Shower	Tumbler	Walking stick	Plate	Pencil	Electric kettle	Pen	Eraser	Earphones
Towel	Chopsticks	Meat	Fish	Hat	Rice	Kimchi	Bread	Cushin	Mattress

## Data Availability

Publicly available datasets were analyzed in this study. COCO dataset can be found here: https://cocodataset.org/ (accessed on 30 September 2021).

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
