# Peer review of "Automatic Object Detection Algorithm-Based Braille Image Generation System for the Recognition of Real-Life Obstacles for Visually Impaired People"

_sensors, 2022, doi:10.3390/s22041601_

Round 1
Reviewer 1 Report
Dear Authors,
All the suggestions you can find in attached file.

Reviewer 2 Report
===== Synopsis:
This study explores a system that translates an RGB image into a braille pad for tactile reading. The system was tested on three images only and a qualitative evaluation is given using 10 participants.
===== General Comments:
I found the study interesting, but it should be explained a bit more, in particular how the evaluation was carried out. Overall it reads well, but it starts too slow and then accelerates to a pace that is too fast - more balance would be better. The sections on image pre/processings are a bit shaky and I will give detailed comments below.
- Abstract contains too much introduction. It should start with line 8 approximately. The rest of the abstract should be more specific. The number 85% and 6 seconds need to be explained better.
- Figure captions should be more explicit.
- Citations should be more precise.
Authors should directly mention the methods they have used. They should not introduce methods that are not applied in the final system.
===== Specific Comments:
- line 22: people with low vision: "low vision" is unclear. I've never heard of this term.
- line 24: 2.2 billion appears to me a lot. That is one third of mankind. Or do you mean millions?
- citation [29]: big yellow teapot. I'm clueless what is being referred to.
- Figure 19: shows histogram smoothing for one channel. And you show only the smoothened image. It would make sense to show the original histogram too. But since you do not use the method in your final system, you might as well drop it.
4.2.1 Image PreProcessing
- citation [32]: unknown journal? I found a 1977 publication in Computer Graphics and Image Processing:Volume 6, Issue 2, April 1977, Pages 184-195.
- Figure 21: increased contrast value in comparison to which Figure? Figure 17?
To clarify: There exist different color transformation models, for example RGB > YCrCb, RGB > LAB. To any of these color spaces one can apply histogram equalization methods. You can apply histogram equalization also to the RGB space.
- Figure 24 is confusing because it does not explain where the three levels are.
4.3.1.Image Processing
You can simplify the explanations by saying that you create a scale space with a Gaussian function and then apply contour detection to each scale. Three scales are generated, with sigmas x, y and z. Those three scales correspond to the three levels of the outline.
[[ After you have applied the Canny algorithm, you obtain a black-white image with ON pixels (value = 1) corresponding to an edge (of an object), and OFF pixels (value = 0) corresponding to background or region. To such a black-white map you then apply morphological operations. ]]
- lines 252-257: 7*7, 5*5, 3*3? Are these neighborhood sizes? If yes, it's probably in pixels. But I think you better specify the sigmas. I assume they are between 1 and 5 somewhere.
- line 263: Say as follows. We apply morphological operations 'erosion' and 'dilation' to remove noise [cite book, such has Digital Image Processing, but not OpenCV].
- line 263: how is the line thickened? I suspect by applying morphological dilation with n=3 (3 repetitions).
4.3 Outline Generation
Better use the word 'map' for the three levels, not image. The outline consists of three maps (levels).
4.4.1. n*n Comparison Conversion
What you do is you convert the maps to braille pads using morphological operations. The type of morphological operation is more complex than erosion or dilation. See morphological operations in the skimage module in Python.
- line 280: say 5x5 pixel neighborhood (not grid)
- Figure 27: makes only sense if you also show an output map next to it, the Braille map for level x.
- lines 326-328: does this include the 10 participants mentioned in the conclusion section?
More explanations are needed.
- line 334: 20 seconds? It's not clear what that includes. You mentioned shorter durations before.
Conclusion: "A test was also conducted on 10 visually impaired people; all participants had a 349
satisfactory response, and it was proposed to simplify the tactile information into a simple 350
shape rather than a complex real object".
The first part of the sentence needs to go to the evaluation section. The second part needs to be better explained - and moved to the introduction.
Round 2
Reviewer 2 Report
Two more issues:
- The last sentence in the abstract (line 20): sounds unclear, because you do not mention the two types. Better: Participants preferred braille maps that contained only the outline of the objects, over braille maps containing the full texture details. (or something similar).
- Canny algorithm: there are typically two parameters: a threshold (actually two), and a sigma parameter. The sigma parameter represents the width of the Gaussian curve (it is the more important parameter of the two). Sigmas are usually specified between 1 and 5. If you have chosen neighborhood sizes of 3x3, 5x5 and 7x7 pixels, I suspect the sigma values were equal 1,2 and 3, respectively (because the larger the sigma, the larger needs to be the neighborhood to be effective). But that depends on what software was used. Specifying 3x3Sigma does not make sense.
